# Constraining Embedding Learning with Self-Matrix Factorization

## Abstract

We focus on the problem of learning object representations from solely association data, that is observed associations between objects of two different types, e.g. movies rated by users. We aim to obtain embeddings encoding object attributes that were not part of the learning process, e.g. movie genres. It has been shown that meaningful representations can be obtained by constraining the learning with manually curated object similarities. We propose Self-Matrix Factorization (SMF), a method that learns object representations and object similarities from observed associations, with the latter constraining the learned representations.[AA] In our extensive evaluation across three real-world datasets, we compared SMF with SLIM, HCCF and NMF obtaining better performance at predicting missing associations as measured by RMSE and precision at top-K. We also show that SMF outperforms the competitors at encoding object attributes as measured by the embedding distances between objects divided into attribute-driven groups.

## 1 Introduction

This paper focuses on the problem of learning object representations from observed associations between objects of two different types. We assume that no data is available other than the associations between the objects, and our aim is to obtain representations that reflect object attributes and properties that, in general, are unknown.

The observed associations between two groups of $n$ and $m$ objects respectively can be represented by a data matrix $X \in \mathbb{R}^{n \times m}$, where the association between object $i$ and object $j$ is stored in $X_{i,j}$. We consider the case in which this value can be either a binary or a positive number, and a value of zero indicates no known association between objects $i$ and $j$.

Learning meaningful object representations has been shown to be relevant for several tasks including recovering missing associations and clustering objects into meaningful groups, and several methods have been proposed. Matrix Factorization (MF) methods, for instance, assume that the association matrix is low rank, allowing $X$ to be decomposed into lower-dimensional matrix factors, containing the representations of the objects. Techniques like principal component analysis (PCA) (Hotelling, 1933), singular value decomposition (SVD)(Eckart & Young, 1936) and non-negative matrix factorization (NMF) (Lee & Seung, 1999) have been successfully applied to tasks such as recovering associations (Sarwar et al., 2002; Vozalis & Margaritis, 2007; Luo et al., 2014) and clustering objects (Yang & Seoighe, 2016; Yeung & Ruzzo, 2001).

Deep Learning is another widely used technique for learning object embeddings, particularly through graph neural networks (GNNs). An association data matrix can be thought of as a bipartite graph where the nodes represent the objects in the data and the links represent the associations between them. GNNs leverage this network structure to extract insights from the encoded graphs. While deep learning methods have been shown to be particularly effective at incorporating prior known object properties (Wu et al., 2022), a number of techniques have also been developed that can use solely association data such as, for example, LightGCN (He et al., 2020), SEAL (Zhang et al., 2021) and HCCF (Xia et al., 2022).

Recently, learning strategies that rely on manually curated similarities between objects have been proposed to constrain embedding learning somehow. For example, Neo-GNNs (Yun et al., 2021)

and BUDDY (Chamberlain et al., 2023) are GNN methods relying on higher-order interactions in the graph. These interactions function as additional node similarity features and were used to enhance link-prediction performance. However, selecting such similarities is not trivial.

In this paper, we argue that object similarities can be learned directly from the data matrix. We rely on the fact that[AA] the objects lie on multiple linear low-dimensional manifolds embedded in a high-dimensional space (Elhamifar & Vidal, 2013). Our matrix decomposition approach, Self-Matrix Factorization (SMF), learns distributed representations while constraining them using learned object similarities. These similarities depend on the manifold structures implicit in the association matrix $X$ and are learned together with the embeddings. In other words, the object similarities, determined by their positions in the manifolds, naturally constrain the object embeddings during the learning. Our method is the first to explore this idea in a matrix factorization model[AA]. In our extensive evaluation across three distinct benchmark datasets, we show that SMF consistently outperforms the competitors at encoding object attributes as measured by the embedding distances between objects divided into attribute-driven groups. We also performed experiments to recover missing values on the different association matrices and show that SMF obtains comparable or better predictions than its competitors.

## 2 RELATED WORKS

MF and GNN techniques encompass numerous methods for learning object representations from association data (Koren et al., 2021; Wu et al., 2022). MF techniques decompose the association matrix $X$ into two or more matrix factors, where the object representations are encoded as rows or columns of these matrix factors, mapping objects to a shared latent space of lower dimensionality (Aggarwal et al., 2016). Several methods for link prediction have been proposed, including SVD (Koren et al., 2009), SVD++ (Koren, 2008) and probabilistic matrix factorization (Yang et al., 2014). NMF (Lee & Seung, 1999) and its variations have been used across fields ranging from medicine to engineering (Hamamoto et al., 2022; Sturluson et al., 2021). Graph-regularized NMF (Cai et al., 2010), symmetric NMF (Luo et al., 2021) and robust NMF (Peng et al., 2021) have been successfully used for object clustering and community detection. Additionally, NMF with l1, l2 or elastic net regularization has been applied successfully across diverse applications, including precision medicine (Hamamoto et al., 2022), gene-expression analysis (Sweeney et al., 2023) and recommender systems (Rendle et al., 2020), showing state-of-the-art performance.[AA]

GNNs have gained popularity for their strong capabilities in graph representation learning. These methods can effectively learn node representations that are well-suited for link prediction tasks (Zhang et al., 2021). One advantage of GNNs is their ability to incorporate external object features, which can significantly enhance prediction performance (Wu et al., 2022). Some approaches, like graph-regularized NMF (Cai et al., 2010), BUDDY (Chamberlain et al., 2023), and Neo-GNNs (Yun et al., 2021), leverage similarity measures to improve object clustering and link prediction performance. HCCF, a specialized GNN technique, learns hyper-edges between objects, enabling it to simultaneously learn embeddings and refine object similarities for improved representation learning.[AA]

Manually curated similarities have proven useful for embedding learning, stemming from the fact that these similarities can themselves be used in recommender systems (Aggarwal et al., 2016). Sparse Linear Models (SLIM) (Ning & Karypis, 2011) are state-of-the-art recommender systems (Ferrari Dacrema et al., 2019) that rely on learning object similarities rather than embeddings. SLIM learns coefficients such that each object can be represented as a linear combination of other objects. This means that a new link between objects $i$ and $j$ is predicted only if objects similar to $i$ were originally linked with $j$. The coefficients used to reconstruct objects depend on the linear manifolds present in the data matrix $X$. In this way, new links are recommended to an object based on the links other objects belonging to the same linear manifold have. Although these similarities have demonstrated predictive power, they have not yet been used to inform embedding learning. In this work, we address this gap by proposing a framework that jointly learns object embeddings and object similarities, where the latter constrains the embedding space, resulting in richer representations.[AA]

# 3 SELF-MATRIX FACTORIZATION

SMF learns two non-negative matrices $W \in \mathbb{R}^{n \times k}$ and $H \in \mathbb{R}^{k \times m}$, with $k << (m \times n)$. Each matrix contains distinct low dimensional object embeddings, such that their product approximates the low-rank interaction data matrix $X \in \mathbb{R}^{n \times m}$:

$$X \simeq WH. \tag{1}$$

While this model is not new, its novelty[AA] resides in the learning of the embeddings in $W$ to encode linear manifold information implicitly contained in the association data itself. Relying on the above mentioned assumption that objects lie on multiple linear low-dimensional manifolds embedded in high-dimensional space (Elhamifar & Vidal, 2013), let us consider the situation depicted in Figure 1a in which we have points in the 3-D space that are approximately localized onto 3 distinct linear manifolds. Rows of $X$ are represented as squares, triangles and circles, with triangles and squares lying on one-dimensional sub-space (red and brown lines) and circles lying on a two-dimensional sub-space (green plane). Let us focus on the three blue points of which $i$ and $p$ lie on the plane and $q$ on the red line.[AA]. We assume that objects that belong to the same subspace, are more similar to each other than objects that reside in different subspaces. We would like these similarities to constrain the learning of the embeddings – that is, we would like the embedding for two objects that belong to the same subspace, to be more similar to each other than the embeddings of objects that reside in different subspaces. Thus, in the embedding space (2-dimensional, in Figure 1b), object $i$ should be closer to object $p$ than to object $q$, mimicking their behavior in the high-dimensional space. Figure 1b demonstrates the expected behavior of SMF-learned object embeddings. Points that belong to the same linear manifold in the high-dimensional space are projected into a lower-dimensional space, where they closely approximate one another.[AA]

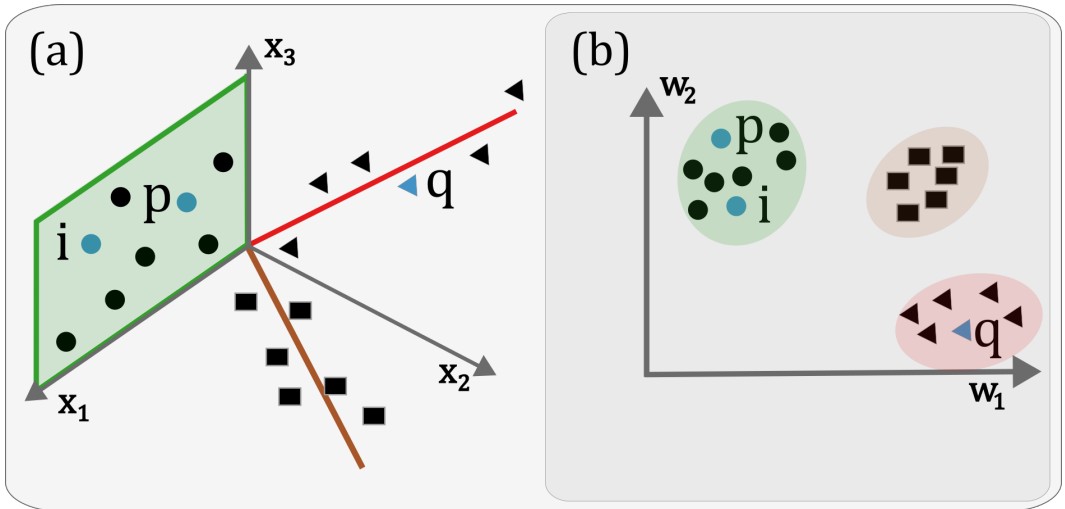

Figure 1: SMF explicit constraint. In this example, the association matrix $X$ contains only 3 columns. $X$ is decomposed into the product $WH$, where $W$ have 2 columns. (a) Positions of $X$ rows in the 3-dimensional space. Points represented as dots, triangles and squares belong to different subspaces. (b) Positions of the 2-dimensional rows of $W$ in the space, SMF uses the similarities established by the linear manifolds to constrain $W$ such that a pair of object embeddings are likely to have a high dot product if they belong to the same linear manifold in the 3-dimensional space.

We propose the following loss function for learning a model with these properties:

$$\min_{W,H} \mathcal{L}_{\text{SMF}}(W, H) = \frac{1}{2}\|X - WH\|_F^2 + \frac{\lambda_{se}}{4}\|X - [T \circ (WW')]X\|_F^2$$

$$+ \lambda_1\|W\|_1 + \lambda_1\|H\|_1 + \frac{\lambda_2}{2}\|W\|_F^2 + \frac{\lambda_2}{2}\|H\|_F^2 \tag{2}$$

$$\text{subject to } W, H \geq 0.$$

where $\circ$ represents the element-wise product and $\|\cdot\|_F^2$ indicates the Frobenius norm. The first term of Equation 2 is the Euclidean distance between the non-negative matrix $X$ and the product of two non-negative matrices $W$ and $H$. Minimizing this distance results in projecting high-dimensional data into a low-dimensional representation. The non-negativity constraint of matrices $W$ and $H$ is a crucial factor for the interpretability of the representations (Lee & Seung, 1999). This constraint naturally encourages any pair of vectors $W_{i,:}$ and $H_{:,j}$ to exhibit a significant overlap in their high-valued components if the objects $i$ and $j$ share an observed association in $X$. Conversely, objects that do not meet this condition can be understood as lacking common attributes, making them less likely to interact.

While parts of Equation 2 resemble the loss function of NMF, its second term introduces a fundamental novelty.[AA] It is designed to preserve the linear manifold information implicit in the matrix $X$. Matrix $T$ is populated with ones, except for the diagonal where elements are set to zero. By reducing the distance between the matrix $[T \circ (WW')]X$ and the original association matrix, we are reconstructing each row of $X$ using other rows of $X$:

$$X_{i,:} = \sum_j T_{i,j}[WW']_{i,j}X_{j,:}$$

Note that, since we are learning $W$, the presence of the matrix $T$ is necessary to avoid the trivial solution in which $WW'$ becomes the identity matrix. The last 4 terms apply elastic-net regularization to the matrices W and H to promote sparsity and mitigate overfitting. Therefore, since minimizing the loss function in Equation 2 is attempting to reconstruct a row using only a few other rows, the learning will favour reconstructing each row using only rows representing the objects in the same subspace. For instance, in Figure 1a one can reconstruct each point in the plane by using only points in the plane, without the need to use points from different subspaces. Let us also note that matrix $[T \circ (WW')]$ contains the coefficients for the reconstruction of the rows of $X$. Therefore, matrix $WW'$ is attempting to encode the inherent similarities between the objects established by the linear manifolds. During the learning of $W$, this amount to promote higher values for the dot product between the lower-dimensional representations of objects within the same subspace than for the dot product between representations of objects in different subspaces.

By minimizing the loss function in equation 2, we approximate each interaction $X_{i,j}$ as $(W_{i,:} \cdot H_{:,j})$ (first term) as well as $\sum_s T_{i,s}(W_{i,:} \cdot W'_{s,:})X_{s,j}$ (second term). The first term enforces shared latent features between the rows and column objects, while the second term incorporates an explicit constraint for all the embeddings of the objects in the row of $X$. This second constraint is directly related to the similarity between object embeddings in $W$, so that the dot product between any pair $W_{i,:}$ and $W_{p,:}$ is informed by the linear manifolds in which objects $i$ and $p$ lies. Notably, SMF does not require prior knowledge of these manifolds; instead, it simultaneously learns the embeddings and the manifold structure, making it the first method to integrate these two processes.[AA]

Similarly to NMF (Lee & Seung, 2000), we derived a multiplicative update rule to minimize the function in Equation 2:[AA]

$$W_{i,j} \leftarrow W_{i,j} \times \frac{[XH' + \lambda_{se}((XX') \circ T)W]_{i,j}}{[WHH' + \lambda_{se}(((T \circ (WW'))XX') \circ T)W + \lambda_2 W + \lambda_1 \text{sgn}(W)]_{i,j}} \tag{3}$$

$$H_{i,j} \leftarrow H_{i,j} \times \frac{[W'X]_{i,j}}{[W'WH + \lambda_2 H + \lambda_1 \text{sgn}(H)]_{i,j}} \tag{4}$$

where $sgn(\cdot)$ is the sign function.[AA] $W$ and $H$ were initialized with non-negative values to ensure that the proposed multiplicative update rules process results in non-negative embeddings after each iteration.

Finally, since in our data matrix $X$ the zeros denote our lack of knowledge about a possible association, it is often convenient to modulate the importance of the zeros during the learning. This has been done by other authors before us (e.g. (Galeano et al., 2020; Blondel et al., 2008)) and is normally achieved by weighting the contribution of the zero values by a factor $\alpha << 1$ in the loss function. In this case, our loss function becomes:

$$\min_{W,H} \mathcal{L}_{\text{WSMF}}(W, H) = \frac{1}{2}\|P \circ (X - WH)\|_F^2 + \frac{\lambda_{se}}{4}\|P \circ (X - [T \circ (WW')]X)\|_F^2$$

$$+ \lambda_1\|W\|_1 + \lambda_1\|H\|_1 + \frac{\lambda_2}{2}\|W\|_F^2 + \frac{\lambda_2}{2}\|H\|_F^2 \tag{5}$$

$$\text{subject to } W, H \geq 0.$$

where the matrix $P \in \mathbb{R}^{n \times m}$ is defined as $P_{i,j} = 1$ when $X_{i,j} > 0$ and $P_{i,j} = \alpha$ otherwise.

Equations 3 and 4 are then modified accordingly to optimize Equation 5:

$$W_{i,j} \leftarrow W_{i,j} \times \frac{[XH' + \lambda_{se}((XX') \circ T)W]_{i,j}}{[((P \circ P) \circ (WH))H' + \lambda_{se}((((P \circ P) \circ ((T \circ (WW'))X))X') \circ T)W + \lambda_2 W + \lambda_1\text{sgn}(W)]_{i,j}} \tag{6}$$

$$H_{i,j} \leftarrow H_{i,j} \times \frac{[W'X]_{i,j}}{[W'((P \circ P) \circ (WH)) + \lambda_2 H + \lambda_1\text{sgn}(H)]_{i,j}} \tag{7}$$

From equations 5, 6, and 7, and assuming that $k << n, m$, it follows that the time complexity of each iteration of SMF optimization algorithm is $O(n^2 \cdot m)$. We discuss the complexity of SMF in the Appendix A.5, together with details about computational time and number of iterations.[AA]

## 4 EXPERIMENTAL RESULTS

To test SMF, we compared its performance against three different models, namely NMF with elastic-net regularization (Pauca et al., 2006), Hypergraph Contrastive Collaborative Filtering (HCCF) (Xia et al., 2022) and Sparse Linear models (SLIM) (Ning & Karypis, 2011). We compared the performance of the models at predicting associations and the quality of the embeddings through clustering analysis. NMF with elastic-net regularization and HCCF were chosen as representative of matrix decomposition techniques and GNN-based methods, respectively. NMF has been shown to be effective at recovering missing associations and encoding object attributes. HCCF is a state-of-the-art GNN model for link prediction. Like SMF, HCCF also learns similarities between objects of the same type, avoiding dependence on manually curated similarities. However, it enhances embeddings by learning hyper-edges (connections involving more than two objects) that contribute to the embedding construction. For the task of predicting associations, we also compared SMF with SLIM (Ning & Karypis, 2011), a state-of-the-art approach for predicting missing associations (Ferrari Dacrema et al., 2019) that has been shown to be competitive with deep learning models. SLIM does not learn object embeddings and for this reason, we could not perform any clustering analysis. Size embeddings for the models are given in the last three columns of Table 1, other algorithm details and their implementation are available in the Appendix, while the code we used in our experiments is available in the Supplementary Material.

We run our experiments on three datasets namely, Movielens, Drug-SE and ModCloth. These datasets were chosen among those that have been used for the task of predicting associations because they also included attribute information of the objects. Overall details are given in Table 1 and descriptions of the three datasets are given below.

**Movielens:** This dataset describes ratings ranging from 1 to 5 that users gave to movies. It is a smaller version of Group lens that is made available for educational and development purposes (Harper & Konstan, 2015). The one used in this work includes object attributes for both users and movies. Each movie is associated with its respective genre (18 genres in total), and each user is

associated with its gender (2 genders in total). The data matrix contains 943 users and 1682 movies. It contains 100000 non-zero elements representing the known ratings, resulting in an association data matrix with a density equal to 6.3%.

**Frequencies of Drug Side Effects (Drug-SE):** Galeano et al. obtained a data matrix containing the frequencies in which certain drugs produce specific side effects(Galeano et al., 2020) by filtering the frequencies obtained from the Side Effect Resource Database (SIDER) (Kuhn et al., 2016). Integers between 1 and 5 represent side-effect frequency terms for 'very rare', 'rare', 'infrequent', 'frequent', and 'very frequent' respectively.

Drugs can be grouped by clinical activity using their main Anatomical, Therapeutic and Chemical (ATC) class levels. ATC is a hierarchical organization of terms maintained by the World Health Organization. A term at a lower level indicates a more specific descriptor of clinical activity. Each drug in the matrix $R$ is associated with its respective ATC-category term in three different levels. The drugs in this dataset belong to all 14 groups at the more general Anatomical level, 70 out of 94 groups at the intermediate Therapeutic level and 147 out of 262 groups at the more specific Chemical level. The data matrix contains 759 drugs and 994 side effects. It contains 37441 non-zero elements representing known frequencies, resulting in an association matrix with a density of 5%.

**ModCloth:** This dataset contains ratings that users gave to different clothing items (Misra et al., 2018). Originally, a rating in this dataset could be 2, 4, 6, 8, or 10. However, we divided all the values by 2, resulting in ratings ranging from 1 to 5. Due to the low density of known ratings in the association matrix, we eliminated those users and clothes with less than 10 associations from the data matrix. The resulting matrix had higher density but still had objects with less than 10 associations. The final data matrix contains 5419 clothing items and 32.089 users with 91900 associations, resulting in a 0.05% density. Each clothing item belongs to 1 of 66 different categories.

Table 1: Datasets and embedding sizes

| Datasets | rows | columns | density | NMF | HCCF | SMF |
|---|---|---|---|---|---|---|
| **Movielens** | 943 | 1682 | 6.3% | 10 | 32 | 10 |
| **Drug-SE** | 759 | 994 | 5% | 10 | 32 | 10 |
| **ModCloth** | 5419 | 32089 | 0.05% | 30 | 32 | 30 |

### 4.1 PERFORMANCE EVALUATION AT PREDICTING ASSOCIATION

**SMF achieves scores closer to the real values.** We evaluated SMF by assessing the model's performance at recovering the different levels of associations. In our experiments, we set 10% of the known associations to zero and then we compared the performance of the different models at recovering them.

We used Root Mean Square Error (RMSE) to assess the reconstruction of the association matrix. The outcomes of the evaluations for NMF, SLIM and SMF are reported in Tables 2 as the mean RMSE across 30 runs of each model, along with the corresponding variance. HCCF is not included because it cannot predict the ratings nor the frequencies, only the presence of a link.

The RMSE is a comparison between known associations and the scores predicted by the models, where lower RMSE values indicate that the predicted scores are closer to the actual associations. We can see that SMF scores remain consistently closer to the real values than those produced by NMF and SLIM. SMF achieves a 15% lower RMSE than NMF in the sparser ModCloth dataset and at least 65% lower RMSE than SLIM for all datasets.

**SMF achieves better performance at top-K predictions.** It is important to measure a system's ability to predict the existence of associations between objects, independently of their specific values. For our datasets, this amounts to predicting which movies is a user more likely to watch, which side effects is a drug likely to cause, and which clothes is a user more likely to rent. To measure this, we generated three binary datasets. These new datasets were built by replacing the non-zero elements on all the datasets. In practice, we are often interested in predicting only a small number of associations with high accuracy. This is commonly referred to as the top-K recommendation task

Table 2: Root Mean Square Error

| MODELS | Movielens | Drug-SE | ModCloth |
|--------|-----------|---------|----------|
| NMF | $0.9777 \pm 3\mathrm{e}{-5}$ | $0.6558 \pm 1\mathrm{e}{-4}$ | $1.5759 \pm 3\mathrm{e}{-4}$ |
| SLIM | $2.9480 \pm 3\mathrm{e}{-6}$ | $1.8622 \pm 5\mathrm{e}{-6}$ | $3.7790 \pm 4\mathrm{e}{-7}$ |
| SMF | $\mathbf{0.9352 \pm 1e{-5}}$ | $\mathbf{0.6455 \pm 5e{-5}}$ | $\mathbf{1.3258 \pm 1e{-5}}$ |

(Cremonesi et al., 2010) where a system's performance is measured using precision at top-K. For this purpose, we ranked the scores to retrieve the K higher elements. These were predicted as new associations between objects and compared with the test set to obtain the precision at top-K, which is the ratio of known associations within the predicted associations. To have fair measurements, the true positives for the analysis are the ones on the test set, and all the unknown elements on the original datasets are considered as true negatives (Krichene & Rendle, 2020). The outcomes of the evaluations are reported in Figure 2, where SMF outperforms the competitors in almost every setting. ModCloth datasets results are not shown due to the low association density, all the models only manage to predict a few true positives at the top 1000.

We can see that SMF predictions achieve the top precision in 7 out of 10 settings. HCCF achieves a better precision at the top 10 and top 20 for the Drug-SE dataset, indicating that it contains more true associations in the top predictions. However, the precision drops as K increments, resulting in the worst precision of all the models for the tops 50, 100 and 150 in the same Drug-SE dataset. SLIM, a state-of-the-art predictor for top-K recommendations (Ferrari Dacrema et al., 2019), achieves the best precision for the top 150 in the Drug-SE dataset.

The Area under the receiver operating characteristic curve (AUROC) and the Area under the precision-recall curve (AUPRC) are also useful metrics to evaluate the overall distribution of the true positives. The AUROC, AUPRC and correlation outcomes for all datasets are shown in the Appendix.

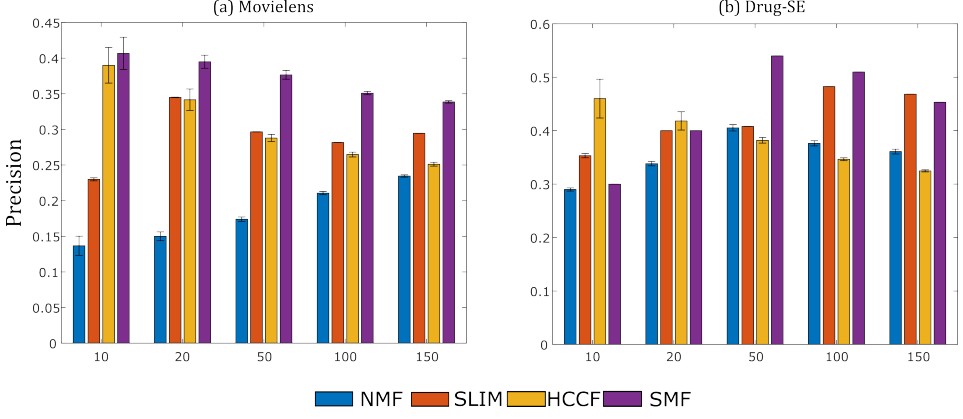

Figure 2: Precision at top-K: Bar plot of the precision of NMF, SLIM, HCCF and SMF for different values of $K$ while predicting missing links in the interaction data. The error bars indicate the variance of the precision for 30 different runs of the models. (a) Precision for the Movielens dataset while predicting links between users and movies, the negative to positive ratio in the test set is approximately 1600. (b) Precision for the Drug-SE dataset while predicting links between side effects and drugs, the negative to positive ratio in the test set is approximately 200.

**SMF sensibility to hyperparameters settings:** The SMF loss function proposed in Eq. 5 contains five hyperparameters. SMF demonstrates stable performance across a wide range of hyperparameter values, indicating that its practical application does not require extensive hyperparameter tuning. The parameter $\lambda_{se}$ controls the importance of the self-expressive term and we set it to 1 in all experiments in this paper. Figure 4 in the Appendix A.4 explores the effect other hyperparameters have on embedding learning by assessing the RMSE and AUPRC on the validation set using the Movie-

Lens dataset. SMF is robust to the choice of the object embedding dimension $k$, achieving good performance even for low values of $k$. As it was also shown by other authors (Galeano et al., 2020), $\alpha$ value depends on the task. $\alpha$ should be set to a low value (closer to zero) when the objective is to accurately retrieve the numerical values of the associations, as in tasks focused on minimizing RMSE. Conversely, $\alpha$ should be set to a high value (closer to 1) when correctly identifying the associations themselves is more critical, as in tasks that optimize AUPRC. Additionally, this experiment shows that SMF is resilient to different values of the $\lambda_1$ and $\lambda_2$ regularization weights. Finally, performance remains consistent across the explored search space, with the only significant variations arising predictably from changes in $\alpha$.[AA]

## 4.2 Embedding evaluation through cluster analysis

To demonstrate that the SMF-derived embeddings offer a more meaningful encoding of previously unseen latent object attributes, we analyzed these low-dimensional representations and compared them with the ones learned by NMF and HCCF. Our aim is to assess their capacity for encapsulating inherent data characteristics, which were not part of the training process but may play crucial roles in establishing connections between objects. Our embedding analysis was conducted on two levels: first, to verify whether SMF effectively clusters objects into meaningful groups within the low-dimensional space; and second, to assess whether SMF achieves superior class separation of objects compared to NMF and HCCF.

We took advantage of this information and grouped the embeddings in $W$ into disjoint sets based on the classes to which their corresponding objects belong. Subsequently, we calculate a similarity matrix, $W_{sim} \in \mathbb{R}^{n \times n}$, containing the cosine similarity between all the embeddings. Finally, we employ a two-sample $t$-test to assess whether the similarities between objects within the same class (intra-similarities) differ significantly from the similarities between objects in different classes (inter-similarities). An illustration of this procedure is provided in Figure 3a. This process was repeated 30 times across different runs of the models.

For the Movielens dataset, we organized users based on their gender, and movies by their respective genres. In our genre analysis, we refined the dataset to include only movies with a single genre, enabling the classification of movies into disjoint categories. In the case of the Drug-SE dataset, we categorized drugs according to their various levels within the ATC hierarchy[1]. The lower levels of the ATC hierarchy provide more specific terms for drug classification. For this study, we compared the similarity of drug embeddings across three levels of the hierarchy: anatomical, therapeutical, and chemical. Finally, the clothing items of the ModCloth dataset can be separated into different groups depending on which type of clothes they are (dresses, jeans, blazers, etc).

**SMF consistently clusters objects in the low-dimensional space.** In our analysis of the distributions of intra- and inter-similarities for the clothing types included in the ModCloth dataset, all NMF, HCCF and SMF achieved significant separation in $100\%$ of the runs. However, for the Movielens dataset, clustering movies by gender in the embedding space proved challenging for HCCF. NMF, HCCF and SMF attain significant separation $100\%$, $67\%$ and $93\%$ of the runs, respectively.

When considering the various levels of the ATC drug classification hierarchy, SMF and HCCF achieved significant distribution separation $100\%$ of the runs for every level. On the other hand, NMF struggled to maintain consistent separation, achieving statistical significance in only $3\%$, $13\%$, and $13\%$ of runs for the $1^{st}$, $2^{nd}$, and $3^{rd}$ levels, respectively. The consistent achievement of statistical significance in these experiments indicates effective clustering of objects in the low-dimensional space. This provides compelling evidence that the SMF-learned embeddings reliably encode meaningful information about the fundamental attributes of the objects.

**SMF achieves superior class separation.** To assess the efficacy of each method in achieving class separation, we employ the Z-score difference between the means of the intra-class and inter-class similarity distributions:

$$Z = \frac{\mu_{in} - \mu_{out}}{\sqrt{\frac{\sigma_{in}^2}{n_{in}} + \frac{\sigma_{out}^2}{n_{out}}}},$$

---

[1] ATC categories were obtained from the ATC codes WHO 2018 release.

where $\mu_{in}$ is the average embedding similarity of object pairs in the same group. $\mu_{out}$ is the average embedding similarity of object pairs in different groups. $\sigma_{in}$ and $\sigma_{out}$ are the corresponding standard deviations, and $n_{in}$ and $n_{out}$ are the corresponding number of object-pairs.

We can interpret the z-score as a normalized distance that measures how different two distributions are by adjusting the difference between the means according to their standard deviation.

Our results for this experiment are summarized in Figures 3b, 3.c and 3.d. We can see that SMF-learned embeddings effectively group objects into more meaningful clusters than those learned by both NMF and HCCF, across all the datasets and diverse groups. Notably, in Figure 3d, we observe that the separation between groups in the ATC levels increases as we delve from the first to the second level of the hierarchy. This reflects the fact that the drug clinical activity becomes more similar as we move to more specific levels.

## 5    CONCLUSION AND DISCUSSION

Many machine learning approaches rely on learning distributed representations able to reflect relevant object attributes. A common strategy to enrich these embeddings is by directly constraining them to follow similarities extracted from side information (Aggarwal et al., 2016). Similarly, one can directly rely on the similarities in the association matrix to guide the embedding learning to better uncover patterns in the data. In this work, we introduced Self-Matrix Decomposition (SMF), a constrained matrix decomposition approach that learns low-dimensional representations by constraining them to rely on object similarities. These similarities depend on linear manifolds implicit in the association data and are learned with the representations.

SMF can decompose a low-rank matrix while preserving its inherent similarities in $WW'$, leveraging the relationships between rows of $X$ that are revealed thanks to the second term in Equation 2. This *Self-Expressive* term learns a coefficient matrix that allows $X$ to be reconstructed by itself, similarly to a Self-Expressive model (Elhamifar & Vidal, 2013). The loss function in Equation 5 facilitates the embeddings to learn better representations and capture the latent attributes, as they allow the embeddings to glean information from objects residing within the same subspace (as depicted in Figure 1).

We conducted experiments to assess whether a set of known object properties could be effectively encoded within the object embeddings. Prior research has also delved into similar investigations, revealing, for instance, that various NMF variants can learn embeddings encoding movie genres (Gomez-Uribe & Hunt, 2015) and drug ATC categories (Galeano et al., 2020). We conducted an in-depth analysis of the similarities among embeddings generated by NMF, HCCF and SMF across multiple runs and diverse groupings. This analysis aimed to ascertain whether objects belonging to the same group consistently clustered together in the low-dimensional space. SMF offers significantly higher stability in learning well-separated embeddings compared to NMF and HCCF. This is evident from the fact that in multiple runs, SMF achieves statistical significance approximately 99% of the time, whereas NMF and HCCF accomplish this feat in only 41% and 87% of the runs, respectively. Furthermore, the experimental results demonstrate that SMF consistently achieves superior class separation in all conducted experiments (depicted in Figure 3). Consequently, SMF can be used to cluster objects in meaningful groups, and an analysis of these groups may help reveal hidden object attributes.

The experimental results in the supervised setting indicate that SMF attains overall better RMSE values compared to NMF and SLIM suggesting that the subspaces encoded in the matrix coefficient of $WW'$ contribute to the learning of more descriptive embeddings for recommendations. While AUROC and AUPRC are commonly used to evaluate classification tasks by showing how true positive samples are ranked when predicting associations, practical effectiveness is better measured by the precision at top-$K$ metric. This metric indicates how close the true positives are to the top of the ranking, which is more relevant for suggesting new associations. In this regard, SMF consistently outperforms NMF, SLIM and HCCF, reinforcing the notion that SMF embeddings adeptly capture the distinctive patterns typically learned by reconstructing the data matrix $X$ directly with $W$ and $H$ and indirectly by constraining $W$.

Although SMF aims at learning from association data, it can be applied to any data represented as a nonnegative matrix. Furthermore, we expect it to be able to integrate extra information. For

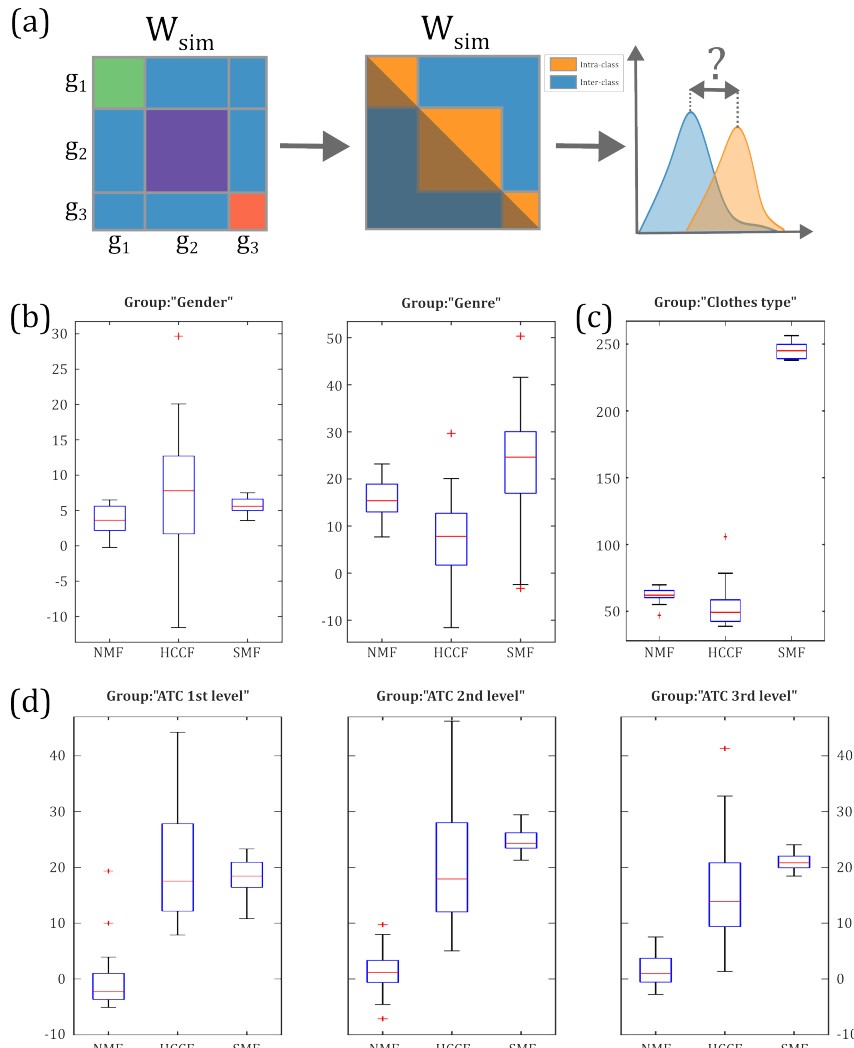

Figure 3: Embedding Analysis: Box plots showing the distributions of the Z-score differences across 30 different runs of NMF, HCCF and SMF. (a) Pipeline explaining the experiment, first, we have the embedding matrix $W$, arranged into three different groups, $g_1$, $g_2$, and $g_3$. Next, $W_{sim}$ contains the similarities between all the embeddings. Lastly, we calculate if there is a statistical significance in the difference between the intra-class and inter-class similarities from $W_{sim}$ (b) Movielens experiments. The left plot shows the separation between the distributions of the similarities while grouping the users by their gender (Male and Female for this dataset). The right plot shows the separation between the distribution of the similarities while grouping the movies by their genres (18 distinct groups). (c) ModCloth experiments. Separation of the distribution of the similarities while grouping the clothes by their type. (d) ATC-category experiments. From left to right, the grouping advances in the ATC hierarchy, $1^{st}$: anatomical, $2^{nd}$: therapeutic, and $3^{rd}$: pharmacological.

instance, to include additional measures of similarity between objects, one could add a term in the cost function that penalizes the difference between the learned similarity and the additional similarity measure.[AA]

Like most machine learning models, SMF does not address biases in the training dataset. As a result, objects with more associations are likely to have higher prediction scores than those with fewer associations. Another limitation of SMF is its inefficiency in handling dynamic datasets. When association data changes, the model must be retrained to ensure that the embeddings reflect the updated state of the objects.[AA]

ACKNOWLEDGMENTS

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

# A  APPENDIX

## A.1  MODELS LEARNING

**SMF**: For our model, we optimized the function shown in Equation 5, using the iterative multiplicative update rule described in Equations 6 and 7 for all the datasets.

**NMF**: Here we opted to train an elastic-net regularized NMF (Pauca et al., 2006), where we modify the loss function similarly as with data-driven regularized NMF (Galeano et al., 2020):

$$\min_{W,H} \mathcal{L}_{\text{WNMF}}(W, H) = \frac{1}{2}\|P \circ (X - WH)\|_F^2$$
$$\text{subject to } W, H \geq 0. \tag{8}$$

that can be optimized with the following multiplicative update rules:

$$W_{i,j} \leftarrow W_{i,j} \times \frac{[(P^2 \circ X)H']_{i,j}}{[(P^2 \circ (WH))H' + \lambda_2 W + \lambda_1 \text{sgn}(W)]_{ij}} \tag{9}$$

$$H_{i,j} \leftarrow Hi, j \times \frac{[W'(P^2 \circ X)]_{ij}}{[W'(P^2(WH)) + \lambda_2 H + \lambda_1 \text{sgn}(H)]_{ij}} \tag{10}$$

**SLIM**: here we modified the original loss by replacing the diagonal constraint by adding the trace of $M$ $(tr(M))$ multiplied by a large number $\gamma$ as a new term in the loss function (Galeano & Paccanaro, 2022). The loss is also modified similarly as in Equation 5 to tune the importance of the zeros during the learning:

$$\min_{M} \mathcal{L}_{\text{WSLIM}}(M) = \frac{1}{2}\|P \circ (X - MX)\|_F^2 + \gamma tr(M) + \frac{\lambda_2}{2}\|M\|_F^2 + \lambda_1\|M\|_1 \tag{11}$$
$$\text{subject to } M \geq 0$$

that can be optimized with the following multiplicative update rules:

$$M_{i,j} \leftarrow M_{i,j} \frac{[(P^2 \circ X)X]_{i,j}}{[(P^2(MX))X' + \gamma I + \lambda_2 M + \lambda_1 \text{sgn}(M)]_{i,j}} \tag{12}$$

All learnable parameters for NMF, SLIM and SMF were initialized by sampling from a uniform distribution between $0$ and $0.1$. It can be proven that in the optimization proposed for Equation 8 and 11, both loss functions are non-increasing at their respective parameters converge in a local minimum for NMF and global minimum for SLIM. For Equation 5, we observed that the loss function always increases for the first iteration, then, for the second iteration forward, the loss function turns out to be non-increasing, and it also manages to achieve convergence for every run of the model, what seems to indicate that this version of adaptive gradient descent is well suited for the problem under study in this work. The algorithm for the optimization of Equation 5 was implemented in Matlab R2023a, and the code is included with this submission. The training is stopped by satisfying the stopping criteria $\delta \leq 1\mathrm{e}{-3}$ for the NMF, SLIM and SMF models, and the maximum relative change $\delta$ is defined as:

$$\delta = \frac{\max(\|W_{i,j}^{\mathrm{old}} - W_{i,j}^{\mathrm{new}}\|)}{\max(\|W_{i,j}^{\mathrm{old}}\|)} \tag{13}$$

where $W^{\mathrm{old}}$ and $W^{\mathrm{new}}$ are the values of the matrix $W$ after each iteration, clearly the same formula is also applied for $H$ and $M$.

**HCCF:** This is a GNN model relying in two message-passing mechanisms. One message-passing occurs between the representation of graph nodes, and the other occurs between hyper-edge representations of the nodes. Both mechanisms are connected by incorporating message and node embeddings generated by the other message-passing, and by the loss function including a contrastive term to connect the two different types of message embeddings:

$$\mathcal{L}_s^{(u)} = \sum_{i=0}^{I} \sum_{l=0}^{L} -log \frac{exp(s(z_{i,l}^{(u)}, \Gamma_{i,l}^{(u)})/\tau)}{\sum_{i'=0}^{I} exp(s(z_{i,l}^{(u)}, \Gamma_{i',l}^{(u)})/\tau)}$$

where, $z_{i,l}^{(u)}$ and $\Gamma_{i,l}^{(u)}$ are the message embeddings for both processes related to the user $i$ for the $l^{th}$) message passing layer. The function $s(.)$ represents a similarity between both embeddings and $\tau$ is a temperature constant that tunes the softmax. The overall loss is:

$$\mathcal{L} = \mathcal{L}_r + \lambda_c(\mathcal{L}_s^{(u)} + \mathcal{L}_s^{(v)}) + \lambda_{wd}\|\Theta\|_F^2 \tag{14}$$

where $\mathcal{L}_r$ is the marginal loss, $\mathcal{L}_s^{(u)}$ and $\mathcal{L}_s^{(v)}$ are the contrastive loss for users and items respectively and the last term is weight decay for the learnable parameters. $\lambda_c$ tunes the importance of the contrastive term in the learning. The final predictions of HCCF are given by:

$$Pr_{i,j} = \Psi_i^{(u)T}\Psi_j^{(v)}$$

where $Psi_i^{(u)}$ and $Psi_j^{(v)}$ are the final embeddings of user $i$ and item $j$. We used the code provided by the authors (Xia et al., 2022) in https://github.com/akaxlh/HCCF to train the HCCF model for our experiments.

## A.2 Implementation Details

Experiments were conducted on a machine equipped with two NVIDIA Quadro RTX 6000 GPUs (each with 24 GB of VRAM), an Intel Xeon Gold 6230 processor, and 192 GB of RAM. SMF, NMF, and SLIM were implemented in MATLAB R2023a, while HCCF was implemented in Python 3.6.12 using TensorFlow 1.14.0. CUDA version 11.8 was utilized to leverage GPU acceleration for HCCF training and evaluation.[AA]

## A.3 AUROC, AUPRC and correlation

Tables 3, 4 and 5 below show extra evaluation metrics used to evaluate the performance of SMF against its competitors. AUROC and AUPRC measure the distribution of true positives in the ranked scores and correlation indicates how close are the predicted scores to the actual levels of association.

Table 3: Results for Movielens dataset

| MODELS | CORRELATION | AUROC | AUPRC |
|---|---|---|---|
| **NMF** | $0.5432 \pm 3-5$ | $\mathbf{0.9441 \pm 1e-7}$ | $0.1402 \pm 5e-6$ |
| **SEM** | $0.3531 \pm 3-7$ | $0.9436 \pm 4e-9$ | $\mathbf{0.1457 \pm 4-8}$ |
| **HCCF** | $(-)$ | $0.8686 \pm 4e-4$ | $0.0611 \pm 2e-4$ |
| **SMF** | $\mathbf{0.5714 \pm 6-6}$ | $0.9436 \pm 2e-7$ | $0.1387 \pm 5-6$ |

Table 4: Results for Drug-SE dataset

| MODELS | CORRELATION | AUROC | AUPRC |
|---|---|---|---|
| **NMF** | $0.7406 \pm 7e-5$ | $0.8819 \pm 4e-10$ | $0.0879 \pm 4e-10$ |
| **SEM** | $0.4181 \pm 4e-7$ | $\mathbf{0.9268 \pm 6e-8}$ | $\mathbf{0.1582 \pm 7e-8}$ |
| **HCCF** | $(-)$ | $0.8113 \pm 3e-5$ | $0.0616 \pm 2e-5$ |
| **SMF** | $\mathbf{0.7432 \pm 1e-5}$ | $0.8582 \pm 2e-8$ | $0.1016 \pm 4e-9$ |

Table 5: Results for ModCloth dataset

| MODELS | CORRELATION | AUROC | AUPRC |
|---|---|---|---|
| **NMF** | $0.0836 \pm 5e-6$ | $0.6904 \pm 3e-5$ | $1.64e-4$ |
| **SEM** | $0.0418 \pm 2e-7$ | $0.6230 \pm 1e-10$ | $2.50e-4$ |
| **HCCF** | $(-)$ | $0.6781 \pm 4e-5$ | $1.97e-4$ |
| **SMF** | $\mathbf{01010 \pm 1e-6}$ | $\mathbf{0.7513 \pm 1e-5}$ | $\mathbf{2.84}e-4$ |

## A.4 HYPERPARAMETER TUNING

A validation set with $10\%$ of the interactions for each dataset was used to select an appropriate set of hyperparameters, the decision was based on RMSE and AUPRC measures. The final set of hyperparameters used to perform the experiments are detailed in Tables 6 and 7.

SLIM does not have a value set for $k$. It is important to note that the value of $\alpha$ the Movielens and ModCloth datasets are only used for the link prediction task (when evaluating precision, AUROC and AUPRC). For rating prediction (when evaluating RMSE and correlation), $\alpha$ is set to zero, reflecting the fact that there are no true zeros in the dataset. Assuming that in an ideal scenario, where all the users assign a rating to all the movies and clothing items, those values should be between 1 and 5. $\lambda_{se}$ in Equation 5 was always set to one.

where $\lambda_{wd}$, $\lambda_c$ and $\tau$ are shown in Equation 14 and $drop$ is dropout. HCCF model training was run with the predefined parameters in https://github.com/akaxlh/HCCF, except for the parameters in Table 7. The set of parameters was selected after multiple rounds of testing against the same validation set used for hyperparameter tuning of NMF, SLIM and SMF.

Figure 4 shows the RMSE (orange) and AUPRC (blue) for different hyperparameter values. The red and grey lines divide the plot into different regions where $k$ (ranging from 4 to 16) and $\alpha$ (ranging from 0 to 1) are constant respectively. Note that within a region where $k$ is constant, there are five regions of constant $\alpha$. Within each of these regions, there are multiple values of the regularization weights in which $\lambda_2$ change after each consecutive point and $\lambda_1$ remains constant for 4 straight points. Both regularization weights range from zero to one and notably the lowest values of AUPRC correspond to regularization weights set to zero. Therefore, SMF generalizes better with regularized embeddings. Here, we illustrated SMF robustness across a wide range of hyperparameter values.[AA]

## A.5 COMPUTATIONAL COMPLEXITY AND SCALABILITY

Table 6: Hyperparameters for Movielens, Drug-SE, and ModCloth

| Values | Movielens | | | Drug-SE | | | ModCloth | | |
|---|---|---|---|---|---|---|---|---|---|
| | NMF | SLIM | SMF | NMF | SLIM | SMF | NMF | SLIM | SMF |
| $\lambda_1$ | 0.5 | 0.5 | 1 | 0.5 | 0.5 | 0.5 | 0.01 | 0.01 | 0.01 |
| $\lambda_2$ | 0.5 | 0.5 | 0 | 0.5 | 0.5 | 0.5 | 0.5 | 1 | 0.5 |
| $\alpha$ | 0.224 | 0.224 | 0.224 | 0.0025 | 0.224 | 0.0025 | 0.05 | 0.05 | 0.05 |

Table 7: Hyperparameters for HCCF

| Values | Movielens | Drug-SE | ModCloth |
|---|---|---|---|
| $\lambda_{wd}$ | $1e-3$ | $1e-2$ | $1e-2$ |
| $\lambda_c$ | $1e-6$ | $1e-7$ | $1e-7$ |
| $\tau$ | 0.1 | 0.1 | 0.1 |
| $drop$ | 0.5 | 0 | 0 |

We derived the computational time complexity of SMF optimization algorithm from equations 5, 6 and 7 that are used to check the convergence criterion and update $W$ and $H$, respectively. As each equation has a fixed number of matrix operations, it suffices to derive the asymptotic complexity of the most expensive one, which involves matrix multiplication. Specifically, the computation of the product between $T \circ (WW')$ and $X$ runs in time $O(n^2 \cdot m)$, where $n$ and $m$ are the dimensions of the data matrix $X \in \mathbb{R}^{n \times m}$. Other multiplications, such as $WH$, run in time $O(n \cdot k \cdot m)$, where $k$ is the dimension of the embedding space. Assuming that $k << n, m$, we have that the time complexity of each iteration of the SMF algorithm is $O(n^2 \cdot m)$.[AA]

SMF iterations have the same time complexity as SEM's ($O(n^2 \cdot m)$), while NMF iterations are less expensive ($O(n \cdot k \cdot m)$). To illustrate the empirical running time of the three approaches, we obtained the mean iteration time (in seconds) in two datasets with different sizes (Movielens and ModCloth). This is shown in the first two columns of Table 8. As SMF's multiplicative update rules involve more matrix multiplications than those of NMF and SEM, it is expected to have a higher execution time per iteration. [AA]

A fundamental variable for analyzing the overall computational time of the SMF algorithm is the total number of iterations until convergence. As it is not possible to derive it analytically, here we show only the empirical total number of iterations. The last two columns of Table 8 show the number of iterations that were necessary to achieve convergence for SMF, SMF, and SEM.[AA]

The scalability of the SMF algorithm is inherently tied to that of NMF, as both share similar computational structures. Like NMF, SMF faces known limitations when applied to large-scale datasets, where the computational demands can hinder efficiency (Gan et al., 2021). This reflects a broader challenge in scaling matrix factorization techniques to accommodate increasingly larger datasets.[AA]

All three methods require additional memory that exceeds the size of the input. Asymptotically, they have the same space complexity as the input $O(n \cdot m)$ when $n \leq m$. When $n >> m$, both SEM and SMF require space $O(n^2)$ to store the similarity matrix.[AA]

Table 8: Mean iteration time and number of iterations

| Models | Movielens (s) | ModCloth (s) | Movielens (it) | ModCloth (it) |
|---|---|---|---|---|
| NMF | 0.01531 | 1.4287 | 1868.20 | 467 |
| SEM | 0.4338 | 7.3948 | 244.20 | 292 |
| SMF | 0.02737 | 7.4693 | 1427.73 | 1032 |

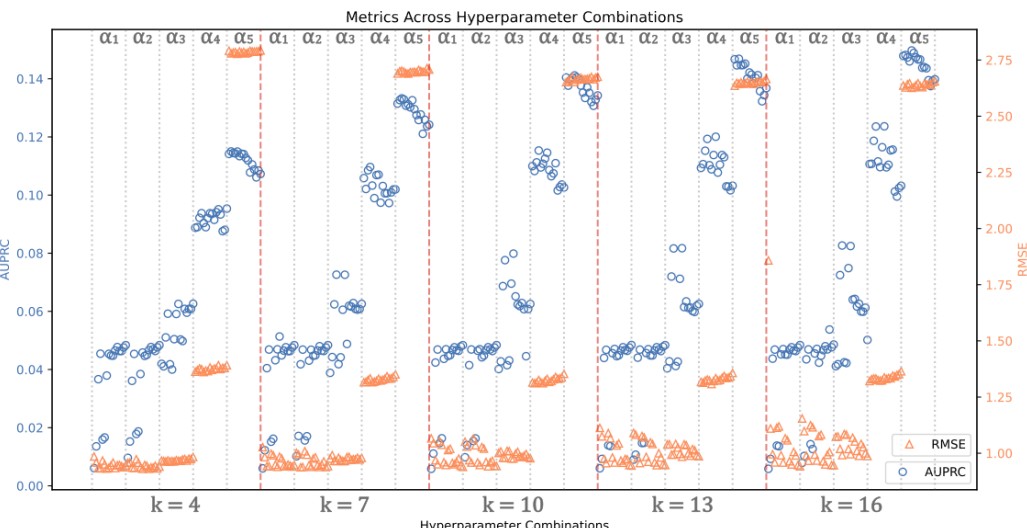

Figure 4: Sensibility to hyperparameter setting: Different points are the RMSE (orange) and AUPRC (blue) of SMF trained with different hyperparameters. Vertical red dotted lines divided the plot into regions where $k$ is constant. Vertical dotted gray lines divided the plot into regions where $\alpha$ is constant ($\alpha_1 = 0$, $\alpha_2 = 0.0025$, $\alpha_3 = 0.05$, $\alpha_4 = 0.22$ and $\alpha_5 = 1$). Within each of these regions, both regularization weights change. $\lambda_2$ is different for each consecutive point, and is set to 0, 0.01, 0.5 and 1 in that order. $\lambda_1$ takes the same values, but it remains the same for 4 straight points before being changed to the next value. The first points at the left of each region correspond to models with both regularization weights set to zero.

