# OpenReview forum: "Constraining embedding learning with Self-Matrix Factorization"
_ICLR.cc/2025/Conference — Submitted to ICLR 2025_

### Official Review · Reviewer_NGpa · 2024-10-31

**Soundness:** 2
**Presentation:** 3
**Contribution:** 1
**Rating:** 5
**Confidence:** 5

**Summary:**

This paper focuses on the problem of learning object representations from solely association data, and proposes a Self-Matrix Factorization (SMF) method. The innovation of this paper is relatively weak, and the core contributions have not been clearly elaborated.

There are several concerns that need to be addressed.

Firstly, the paper relies on the assumption that objects reside on multiple linear low-dimensional manifolds embedded within a high-dimensional space. However, this assumption appears to have already been utilized by numerous prior matrix factorization works, rendering it relatively uninnovative.

Secondly, the paper asserts that object similarities can be derived directly from the data matrix, yet it fails to elucidate the method of learning or the criteria for determining these similarities.

Thirdly, the paper compares its proposed SMF to other methods such as SLIM, HCCF, and NMF, but it does not provide a comprehensive analysis of the strengths and weaknesses of each method.

Forthly, the experiments conducted in this paper are relatively simplistic, both in terms of the datasets and tasks employed, and the comparative methods utilized are outdated.

**Strengths:**

1. This paper focuses on the problem of learning object representations from solely association data, and proposes a Self-Matrix Factorization (SMF) method.
2. The authors performed experiments at recovering missing values on the different association matrices and show that SMF obtains comparable or better predictions than its competitors.

**Weaknesses:**

1. The paper relies on the assumption that objects reside on multiple linear low-dimensional manifolds embedded within a high-dimensional space. However, this assumption appears to have already been utilized by numerous prior matrix factorization works, rendering it relatively uninnovative.

2. The paper asserts that object similarities can be derived directly from the data matrix, yet it fails to elucidate the method of learning or the criteria for determining these similarities.

3. The paper compares its proposed SMF to other methods such as SLIM, HCCF, and NMF, but it does not provide a comprehensive analysis of the strengths and weaknesses of each method.

4. The experiments conducted in this paper are relatively simplistic, both in terms of the datasets and tasks employed, and the comparative methods utilized are outdated.

**Questions:**

1. The paper relies on the assumption that objects reside on multiple linear low-dimensional manifolds embedded within a high-dimensional space. However, this assumption appears to have already been utilized by numerous prior matrix factorization works, rendering it relatively uninnovative.

2. The paper asserts that object similarities can be derived directly from the data matrix, yet it fails to elucidate the method of learning or the criteria for determining these similarities.

3. The paper compares its proposed SMF to other methods such as SLIM, HCCF, and NMF, but it does not provide a comprehensive analysis of the strengths and weaknesses of each method.

4. The experiments conducted in this paper are relatively simplistic, both in terms of the datasets and tasks employed, and the comparative methods utilized are outdated.

---

> ### Author Response · Authors · 2024-11-26
>
> We thank the reviewer for the detailed feedback on our manuscript. Below, we address your comments and show the revisions made to improve the quality of the paper. We have also submitted a new version of the paper, where the changes are highlighted in red and AA stands for anonymous author.
>
> 1. **Model assumption is not new**
>
> We agree that the model assumption is not new. Our novelty is that we are the first to use the linear manifolds that are inherent to the data-matrix to learn object embeddings in a matrix factorization model.
>
> We clarified the fact that the assumption is not new and SMF's innovations in the Introduction, in the new section "Related works", and in the "Self-Matrix Factorization" section.
>
> In the revised version of the introduction, we explained that we rely on a known assumption:
>
> >In this paper, we argue that object similarities can be learned directly from the data matrix. We rely on the fact that the objects lie on multiple linear low-dimensional manifolds embedded in a high-dimensional space (Elhamifar & Vidal, 2013).
>
> We also highlighted SMF's main novelty:
>
> >Our matrix decomposition approach, Self-Matrix Factorization (SMF), learns distributed representations while constraining them using learned object similarities. These similarities depend on the manifold structures implicit in the association matrix $X$ and are learned together with the embeddings. In other words, the object similarities, determined by their positions in the manifolds, naturally constrain the object embeddings during the learning. **Our method is the first to explore this idea in a matrix factorization model**.
>
> In Related Works, we wrote:
>
> > SLIM learns coefficients such that each object can be represented as a linear combination of other objects. This means that a new link between objects $i$ and $j$ is predicted only if objects similar to $i$ were originally linked with $j$. The coefficients used to reconstruct objects depend on the linear manifolds present in the data matrix $X$. In this way, new links are recommended to an object based on the links other objects belonging to the same linear manifold have. Although these similarities have demonstrated predictive power, they have not yet been used to inform embedding learning. **In this work, we address this gap by proposing a framework that jointly learns object embeddings and object similarities, where the latter constrains the embedding space, resulting in richer representations.**
>
> In Self-Matrix Factorization section, we wrote:
>
> > its novelty resides in the learning of the embeddings in $W$ to encode linear manifold information implicitly contained in the association data itself.

---

> ### Author Response · Authors · 2024-11-26
>
> 2. **The paper asserts that object similarities can be derived directly from the data matrix, yet it fails to elucidate the method of learning or the criteria for determining these similarities.**
>
> We explain better our notion of similarity and how our method works in "Related Works" and  "Self-Matrix Factorization section".
> In "Related Works", we explained the notion of similarity from SLIM, which inspires our model:
>
> > SLIM learns coefficients such that each object can be represented as a linear combination of other objects. This means that a new link between objects $i$ and $j$ is predicted only if objects similar to $i$ were originally linked with $j$. **The coefficients used to reconstruct objects depend on the linear manifolds present in the data matrix $X$**. In this way, new links are recommended to an object based on the links other objects belonging to the same linear manifold have. Although these similarities have demonstrated predictive power, they have not yet been used to inform embedding learning. **In this work, we address this gap by proposing a framework that jointly learns object embeddings and object similarities, where the latter constrains the embedding space, resulting in richer representations.**
>
> In "Self-Matrix Factorization", we modified the explanation of the model:
>
> >SMF learns two non-negative matrices $W \in \mathbb{R}^{n \times k}$ and $H \in \mathbb{R}^{k \times m}$, with $k<<(m \times n)$. Each matrix contains distinct low dimensional object embeddings, such that their product approximates the low-rank interaction data matrix $X \in \mathbb{R}^{n \times m}$:
> >$X \simeq WH.$
> >
> >While this model is not new, its novelty resides in the learning of the embeddings in $W$ to encode linear manifold information implicitly contained in the association data itself. Relying on the above mentioned assumption that objects lie on multiple linear low-dimensional manifolds embedded in high-dimensional space (Elhamifar & Vidal, 2013), let us consider the situation depicted in Figure 1.a in which we have points in the 3-D space that are approximately localized onto 3 distinct linear manifolds. Rows of $X$ are represented as squares, triangles and circles, with triangles and squares lying on one-dimensional sub-space (red and brown lines) and circles lying on a two-dimensional sub-space (green plane). Let us focus on the three blue points of which $i$ and $p$ lie on the plane and $q$ on the red line. We assume that objects that belong to the same subspace, are more similar to each other than objects that reside in different subspaces. We would like these similarities to constrain the learning of the embeddings – that is, we would like the embedding for two objects that belong to the same subspace, to be more similar to each other than the embeddings of objects that reside in different subspaces. Thus, in the embedding space (2-dimensional, in Figure 1.b), object $i$ should be closer to object $p$ than to object $q$, mimicking their behavior in the high-dimensional space. Figure 1.b demonstrates the expected behavior of SMF-learned object embeddings. Points that belong to the same linear manifold in the high-dimensional space are projected into a lower-dimensional space, where they closely approximate one another.
>
>  Later we explain the term in the cost function that constrains $W$ so that it will preserve the similarities from the original data:
>
> > While parts of Equation 2 resemble the loss function of NMF, its second term introduces a fundamental novelty. It is designed to preserve the linear manifold information implicit in the matrix.
>
> > By minimizing the loss function in equation 2, we approximate each interaction $X_{i,j}$ as $(W_{i,:}\cdot H_{:,j})$ (first term) as well as $\sum_s T_{i,s} (W_{i,:}\cdot W_{s,:}')X_{s,j}$ (second term). The first term enforces shared latent features between the rows and column objects, while the second term incorporates an explicit constraint for all the embeddings of the objects in the row of $X$. This second constraint is directly related to the similarity between object embeddings in $W$, so that the dot product between any pair $W_{i,:}$ and $W_{p,:}$ is informed by the linear manifolds in which objects $i$ and $p$ lies. Notably, SMF does not require prior knowledge of these manifolds; instead, it simultaneously learns the embeddings and the manifold structure, making it the first method to integrate these two processes.

---

> ### Author Response · Authors · 2024-11-26
>
> 3. **The paper compares its proposed SMF to other methods such as SLIM, HCCF, and NMF, but it does not provide a comprehensive analysis of the strengths and weaknesses of each method.**
>
> In the revised version of the manuscript, we created a section "Related works" where we explain the strengths and weaknesses of each method:
>
> > MF and GNN techniques encompass numerous methods for learning object representations from association data (Koren et al., 2021; Wu et al., 2022). MF techniques decompose the association matrix X into two or more matrix factors, where the object representations are encoded as rows or columns of these matrix factors, mapping objects to a shared latent space of lower dimensionality (Aggarwal et al., 2016). Several methods for link prediction have been proposed, including SVD (Koren et al., 2009), SVD++ (Koren, 2008) and probabilistic matrix factorization (Yang et al., 2014). NMF (Lee & Seung, 1999) and its variations have been used across fields ranging from medicine to engineering (Hamamoto et al., 2022; Sturluson et al., 2021). Graph-regularized NMF (Cai et al., 2010), symmetric NMF (Luo et al., 2021) and robust NMF (Peng et al., 2021) have been successfully used for object clustering and community detection. Additionally, NMF with l1, l2 or elastic net regularization has been applied successfully across diverse applications, including precision medicine (Hamamoto et al., 2022), gene-expression analysis (Sweeney et al., 2023) and recommender systems (Rendle et al., 2020), showing state-of-the-art performance.
> >
> >GNNs have gained popularity for their strong capabilities in graph representation learning. These methods can effectively learn node representations that are well-suited for link prediction tasks (Zhang et al., 2021). One advantage of GNNs is their ability to incorporate external object features, which can significantly enhance prediction performance (Wu et al., 2022). Some approaches, like graph-regularized NMF (Cai et al., 2010), BUDDY (Chamberlain et al., 2023), and Neo-GNNs (Yun et al., 2021), leverage similarity measures to improve object clustering and link prediction performance. HCCF, a specialized GNN technique, learns hyper-edges between objects, enabling it to simultaneously learn embeddings and refine object similarities for improved representation learning.
> >
> > Manually curated similarities have proven useful for embedding learning, stemming from the fact that these similarities can themselves be used in recommender systems (Aggarwal et al., 2016). Sparse Linear Models (SLIM) (Ning & Karypis, 2011) are state-of-the-art recommender systems (Ferrari Dacrema et al., 2019) that rely on learning object similarities rather than embeddings. SLIM learns coefficients such that each object can be represented as a linear combination of other objects. This means that a new link between objects $i$ and $j$ is predicted only if objects similar to $i$ were originally linked with $j$. The coefficients used to reconstruct objects depend on the linear manifolds present in the data matrix $X$. In this way, new links are recommended to an object based on the links other objects belonging to the same linear manifold have. **Although these similarities have demonstrated predictive power, they have not yet been used to inform embedding learning. In this work, we address this gap by proposing a framework that jointly learns object embeddings and object similarities, where the latter constrains the embedding space, resulting in richer representations.**

---

> ### Author Response · Authors · 2024-11-26
>
> 4. **The experiments conducted in this paper are relatively simplistic, both in terms of the datasets and tasks employed, and the comparative methods utilized are outdated.**
>
> In this revised version of the manuscript, we highlighted that the selected comparative methods are a good representative of the start-of-the-art. In the Related Works section, we have included references that justify our choice of competitors.
>
> The selected datasets have the characteristics that suited our analysis: firstly, the associations have different numerical values either than just zero or one. Secondly, the datasets included object attributes other than the associations. These datasets are currently being used as baselines to evaluate different tasks [1][2][3][4][5][6].
>
> **References**
>
> [1] Bao, K., Zhang, J., Zhang, Y., Wang, W., Feng, F., & He, X. (2023, September). Tallrec: An effective and efficient tuning framework to align large language model with recommendation. In Proceedings of the 17th ACM Conference on Recommender Systems (pp. 1007-1014).
>
> [2] Zhang, A., Chen, Y., Sheng, L., Wang, X., & Chua, T. S. (2024, July). On generative agents in recommendation. In Proceedings of the 47th international ACM SIGIR conference on research and development in Information Retrieval (pp. 1807-1817).
>
> [3] Boratto, L., Fabbri, F., Fenu, G., Marras, M., & Medda, G. (2024, October). Fair Augmentation for Graph Collaborative Filtering. In Proceedings of the 18th ACM Conference on Recommender Systems (pp. 158-168).
>
> [4] Zhang, X., Shi, T., Xu, J., Dong, Z., & Wen, J. R. (2024). Model-Agnostic Causal Embedding Learning for Counterfactually Group-Fair Recommendation. IEEE Transactions on Knowledge and Data Engineering.
>
> [5] Liu, W., Zhang, J., Qiao, G., Bian, J., Dong, B., & Li, Y. (2024). HMMF: a hybrid multi-modal fusion framework for predicting drug side effect frequencies. BMC bioinformatics, 25(1), 196.
>
> [6] Xu, X., Yue, L., Li, B., Liu, Y., Wang, Y., Zhang, W., & Wang, L. (2022). DSGAT: predicting frequencies of drug side effects by graph attention networks. Briefings in Bioinformatics, 23(2), bbab586.

---

> > ### Comment · Reviewer_NGpa · 2024-11-27
> > **Response to Rebuttal**
> >
> > I appreciate the author's further elaboration on model assumptions and object similarities in the revision, which has prompted me to raise my score to 5 (borderline reject). However, I still believe that the model assumptions, derived from prior work rather than this study, limit the novelty of the paper. Moreover, the author's claim that the selected comparative methods (including SLIM, HCCF, and NMF) are a good representation of the state-of-the-art is not entirely convincing to me. Furthermore, the overall performance of the proposed method does not outperform the other comparison methods, as evidenced by Tables 3 and 4.

---

> > > ### Author Response · Authors · 2024-11-28
> > >
> > > We thank the reviewer for the valuable feedback and for raising the score. Below we address your comments.
> > >
> > > **However, I still believe that the model assumptions, derived from prior work rather than this study, limit the novelty of the paper.**
> > >
> > > We recognize that the model assumptions are not new, and in the paper we don’t claim that they are. What is new in SMF is: (i) the use of similarities that derived from linear manifolds to constraint the learning, that results in richer embeddings; and (ii) the fact that these similarities are learned together with the embeddings.
> > >
> > > **Moreover, the author's claim that the selected comparative methods (including SLIM, HCCF, and NMF) are a good representation of the state-of-the-art is not entirely convincing to me.**
> > >
> > > We have been searching the literature but we could not find a better choices for the competitors. We are looking for methods that use only association data, and there are not that many. But maybe we missed some – does the reviewer have some suggestions here? It would be much appreciated.
> > >
> > > **Furthermore, the overall performance of the proposed method does not outperform the other comparison methods, as evidenced by Tables 3 and 4.**
> > >
> > > We acknowledge that Tables 3 and 4 demonstrate that SMF performs roughly on par with NMF and SEM. However, these metrics should be considered alongside the results presented in Table 2, Table 5, Figure 2, and Figure 3. Together, these results support our claim that the embeddings learned by SMF are more meaningful. Specifically, SMF is comparable to other methods for certain tasks while outperforming them in others.

---

### Official Review · Reviewer_cSXs · 2024-10-31

**Soundness:** 2
**Presentation:** 1
**Contribution:** 1
**Rating:** 3
**Confidence:** 2

**Summary:**

This paper introduces Self-Matrix Factorization (SMF), a matrix decomposition method that constrains the nonnegative matrix factorization optimization, among other with a "Self-Expressivity" term that aims to preserve the linear manifold information implicit in the original association matrix.

Tested on datasets like MovieLens and Drug-SE, SMF outperformed traditional methods in predicting associations and clustering objects based on latent features (e.g., genres or categories). This method shows promise for recommendation systems and unsupervised learning tasks where labeled data is limited.

**Strengths:**

- The paper explores an interesting topic, ie to generate embeddings that capture implicit object attributes by leveraging similarities inferred from associations

- The addition of the term that exploits the fact that objects (amy) lie on multiple linear manifolds, is interesting and seems to provide some gains over NMF.

**Weaknesses:**

W1) There is no related works section, and the contribution and relationships to the closest matrix factorization methods is unclear. Although a popular topic, there is only a handful of matrix factorization works cited. What are the closest matrix factorization works and how does Eq 2 compares? The second term in Eq. 2 allows each row to be reconstructed from others. Is this the first use of this "self-expressive" constraint in MF and representation learning, or have similar constraints been applied in other methods?
I think that authors should consider adding a dedicated related work section comparing SMF to other recent matrix factorization methods, particularly those using similar self-expressive constraints.

W2) The update rule in Eqs 3-4 are derived from Lee & Seung, 2000 and applied to Eq 2. Unclear if there is any substancial contribution there. Same as the addition of factor alpha that is borrowed from related work.



W3) Figure 1 seems way too generic and fails to adequately illustrate the novel aspects of SMF. Figure 1(a) depicts a generic matrix factorization, which does not highlight SMF’s unique contributions. Figure 1(b) shows linear subspaces, but it lacks clarity on how the method effectively utilizes only points within the same subspace to reconstruct an object.
The authors should consider adding a visual representation of how SMF utilizes points within the same subspace for reconstruction, or including a side-by-side comparison with traditional matrix factorization to highlight SMF's unique approach.

W4) The datasets used for evaluating SMF are relatively small, which limits the generalizability of the results, and the comparative analysis is not extensive. The main competitor in Table 2 is NMF, with modest improvements in RMSE observed for SMF. Additionally, SLIM performs significantly worse than NMF, so it may be more insightful to reorder the rows in Table 2 to better highlight SMF's performance against the second-best model.

**Questions:**

Please see weaknesses above. My main questions are with respect to the differences of this approach to other MF works. Section 4 kind of wraps this up but doesnt discuss relations and what this method offers.

Q1) What would you say are the contributions of this method compared to the closest ones?
Q2) could  you clarify what specific innovations, if any, have been made in deriving these update rules in Eq3-4 compared to previous work? Could you discuss how the incorporation of the alpha factor contributes to the overall novelty of their approach?
Q3) could you include more state-of-the-art matrix factorization methods in the comparative analysis? This would help provide a more comprehensive evaluation of SMF's performance.

---

> ### Author Response · Authors · 2024-11-26
>
> We thank you for your thoughtful comments, which have significantly helped to improve our manuscript. Below we address each comment. We have also submitted a new version of the paper, where the changes are highlighted in red and AA stands for anonymous author.
>
> 1. **Related works**
>
> Following the reviewer's suggestion, we created Section 2 "Related works", where we describe the state-of-the art and explain SMF's main novelties:
>
> > MF and GNN techniques encompass numerous methods for learning object representations from association data (Koren et al., 2021; Wu et al., 2022). MF techniques decompose the association matrix X into two or more matrix factors, where the object representations are encoded as rows or columns of these matrix factors, mapping objects to a shared latent space of lower dimensionality (Aggarwal et al., 2016). Several methods for link prediction have been proposed, including SVD (Koren et al., 2009), SVD++ (Koren, 2008) and probabilistic matrix factorization (Yang et al., 2014). NMF (Lee & Seung, 1999) and its variations have been used across fields ranging from medicine to engineering (Hamamoto et al., 2022; Sturluson et al., 2021). Graph-regularized NMF (Cai et al., 2010), symmetric NMF (Luo et al., 2021) and robust NMF (Peng et al., 2021) have been successfully used for object clustering and community detection. Additionally, NMF with l1, l2 or elastic net regularization has been applied successfully across diverse applications, including precision medicine (Hamamoto et al., 2022), gene-expression analysis (Sweeney et al., 2023) and recommender systems (Rendle et al., 2020), showing state-of-the-art performance.
> >
> >GNNs have gained popularity for their strong capabilities in graph representation learning. These methods can effectively learn node representations that are well-suited for link prediction tasks (Zhang et al., 2021). One advantage of GNNs is their ability to incorporate external object features, which can significantly enhance prediction performance (Wu et al., 2022). Some approaches, like graph-regularized NMF (Cai et al., 2010), BUDDY (Chamberlain et al., 2023), and Neo-GNNs (Yun et al., 2021), leverage similarity measures to improve object clustering and link prediction performance. HCCF, a specialized GNN technique, learns hyper-edges between objects, enabling it to simultaneously learn embeddings and refine object similarities for improved representation learning.
> >
> > Manually curated similarities have proven useful for embedding learning, stemming from the fact that these similarities can themselves be used in recommender systems (Aggarwal et al., 2016). Sparse Linear Models (SLIM) (Ning & Karypis, 2011) are state-of-the-art recommender systems (Ferrari Dacrema et al., 2019) that rely on learning object similarities rather than embeddings. SLIM learns coefficients such that each object can be represented as a linear combination of other objects. This means that a new link between objects $i$ and $j$ is predicted only if objects similar to $i$ were originally linked with $j$. The coefficients used to reconstruct objects depend on the linear manifolds present in the data matrix $X$. In this way, new links are recommended to an object based on the links other objects belonging to the same linear manifold have. **Although these similarities have demonstrated predictive power, they have not yet been used to inform embedding learning. In this work, we address this gap by proposing a framework that jointly learns object embeddings and object similarities, where the latter constrains the embedding space, resulting in richer representations.**

---

> ### Author Response · Authors · 2024-11-26
>
> 2. **What would you say are the contributions of this method compared to the closest ones?**
>
> We changed different parts of the manuscript to clarify the main contributions of SMF. Our novelty is that we are the first to use the linear manifolds that are inherent to the data-matrix to learn object embeddings in a matrix factorization model, and those linear manifolds are learned together with the representations.
>
> In Introduction, we wrote:
>
> >Our matrix decomposition approach, Self-Matrix Factorization (SMF), learns distributed representations while constraining them using learned object similarities. These similarities depend on the manifold structures implicit in the association matrix $X$ and are learned together with the embeddings. In other words, the object similarities, determined by their positions in the manifolds, naturally constrain the object embeddings during the learning. **Our method is the first to explore this idea in a matrix factorization model**.
>
> In Related Works, we wrote:
>
> > SLIM learns coefficients such that each object can be represented as a linear combination of other objects. This means that a new link between objects $i$ and $j$ is predicted only if objects similar to $i$ were originally linked with $j$. The coefficients used to reconstruct objects depend on the linear manifolds present in the data matrix $X$. In this way, new links are recommended to an object based on the links other objects belonging to the same linear manifold have. Although these similarities have demonstrated predictive power, they have not yet been used to inform embedding learning. In this work, we address this gap by proposing a framework that jointly learns object embeddings and object similarities, where the latter constrains the embedding space, resulting in richer representations.
>
> In Self-Matrix Factorization section, we wrote:
>
> > its novelty resides in the learning of the embeddings in $W$ to encode linear manifold information implicitly contained in the association data itself.
> > By minimizing the loss function in equation 2, we approximate each interaction $X_{i,j}$ as $(W_{i,:}\cdot H_{:,j})$ (first term) as well as $\sum_s T_{i,s} (W_{i,:}\cdot W_{s,:}')X_{s,j}$ (second term). The first term enforces shared latent features between the rows and column objects, while the second term incorporates an explicit constraint for all the embeddings of the objects in the row of $X$. This second constraint is directly related to the similarity between object embeddings in $W$, so that the dot product between any pair $W_{i,:}$ and $W_{p,:}$ is informed by the linear manifolds in which objects $i$ and $p$ lies. Notably, SMF does not require prior knowledge of these manifolds; instead, it simultaneously learns the embeddings and the manifold structure, making it the first method to integrate these two processes.

---

> ### Author Response · Authors · 2024-11-26
>
> 3. **What are the closest matrix factorization works and how does Eq 2 compares? The second term in Eq. 2 allows each row to be reconstructed from others. Is this the first use of this "self-expressive" constraint in MF and representation learning, or have similar constraints been applied in other methods?**
>
> Yes, this is the first use of this self-expressive constraint in MF and representation learning. We explain how Equation 2 compares to other models in Section 3 "Self-Matrix Factorization":
>
> >While parts of Equation 2  resemble the loss function of NMF, its second term introduces a fundamental novelty. It is designed to preserve the linear manifold information implicit in the matrix $X$.
>
> > By minimizing the loss function in equation 2, we approximate each interaction $X_{i,j}$ as $(W_{i,:}\cdot H_{:,j})$ (first term) as well as $\sum_s T_{i,s} (W_{i,:}\cdot W_{s,:}')X_{s,j}$ (second term). The first term enforces shared latent features between the rows and column objects, while the second term incorporates an explicit constraint for all the embeddings of the objects in the row of $X$. This second constraint is directly related to the similarity between object embeddings in $W$, so that the dot product between any pair $W_{i,:}$ and $W_{p,:}$ is informed by the linear manifolds in which objects $i$ and $p$ lies. Notably, SMF does not require prior knowledge of these manifolds; instead, it simultaneously learns the embeddings and the manifold structure, making it the first method to integrate these two processes.
>
> 4. **Could you clarify what specific innovations, if any, have been made in deriving these update rules in Eq3-4 compared to previous work? Could you discuss how the incorporation of the alpha factor contributes to the overall novelty of their approach?**
>
> The derivation of the update rules in Eq 3 and 4 follows the same procedure as Lee and Seung. Notice that the innovation of SMF does not rely on how we minimize the cost function. SMF's fundamental novelty is the inclusion of the second term in equation 2 that constrains the learning in a meaningful way.
>
> We modified section 3, page 4, to further clarify that the derivation procedure is not new:
>
> >Similarly to NMF (Lee & Seung, 1999), we derived a multiplicative update rule to minimize the function in Equation 2.

---

> ### Author Response · Authors · 2024-11-26
>
> 5. **Figure 1 seems way too generic and fails to adequately illustrate the novel aspects of SMF.**
> We improved Figure 1 to better illustrate SMF.
>
> On page 3, we improved the model explanation based on Figure 1:
>
> >SMF learns two non-negative matrices $W \in \mathbb{R}^{n \times k}$ and $H \in \mathbb{R}^{k \times m}$, with $k<<(m \times n)$. Each matrix contains distinct low dimensional object embeddings, such that their product approximates the low-rank interaction data matrix $X \in \mathbb{R}^{n \times m}$:
> >$X \simeq WH.$
> >
> >While this model is not new, its novelty resides in the learning of the embeddings in $W$ to encode linear manifold information implicitly contained in the association data itself. Relying on the above mentioned assumption that objects lie on multiple linear low-dimensional manifolds embedded in high-dimensional space (Elhamifar & Vidal, 2013), let us consider the situation depicted in Figure 1.a in which we have points in the 3-D space that are approximately localized onto 3 distinct linear manifolds. Rows of $X$ are represented as squares, triangles and circles, with triangles and squares lying on one-dimensional sub-space (red and brown lines) and circles lying on a two-dimensional sub-space (green plane). Let us focus on the three blue points of which $i$ and $p$ lie on the plane and $q$ on the red line. We assume that objects that belong to the same subspace, are more similar to each other than objects that reside in different subspaces. We would like these similarities to constrain the learning of the embeddings – that is, we would like the embedding for two objects that belong to the same subspace, to be more similar to each other than the embeddings of objects that reside in different subspaces. Thus, in the embedding space (2-dimensional, in Figure 1.b), object $i$ should be closer to object $p$ than to object $q$, mimicking their behavior in the high-dimensional space. Figure 1.b demonstrates the expected behavior of SMF-learned object embeddings. Points that belong to the same linear manifold in the high-dimensional space are projected into a lower-dimensional space, where they closely approximate one another.
>
> We added the following caption to Figure 1:
>
> > SMF explicit constraint. In this example, the association matrix \(X\) contains only 3 columns. \(X\) is decomposed into the product \(WH\), where \(W\) have 2 columns. (a) Positions of \(X\) rows in the 3-dimensional space. Points represented as dots, triangles and squares belong to different subspaces.  (b) Positions of the 2-dimensional rows of \(W\) in the space, SMF uses the similarities established by the linear manifolds to constrain \(W\) such that a pair of object embeddings are likely to have a high dot product if they belong to the same linear manifold in the 3-dimensional space.
>
> 6. **Could you include more state-of-the-art matrix factorization methods in the comparative analysis? This would help provide a more comprehensive evaluation of SMF's performance.**
>
> In this revised version of the manuscript we highlighted that the selected comparative methods are a good representative of the start-of-the-art. In the Related Works section, we have included references that justify our choice of competitors.

---

### Official Review · Reviewer_wLXm · 2024-11-06

**Soundness:** 2
**Presentation:** 2
**Contribution:** 2
**Rating:** 5
**Confidence:** 4

**Summary:**

This paper presents a method, Self-Matrix Factorization (SMF), for learning object representations from association data without prior knowledge of object attributes. The paper claims that SMF outperforms other methods like SLIM, HCCF, and NMF in predicting missing associations and encoding object attributes.

**Strengths:**

- Performance Evaluation: The paper uses a variety of metrics (RMSE, precision at top-K, AUROC, AUPRC) across different datasets to evaluate the model's performance, which provides an assessment of its capabilities.
- Comparison with State-of-the-Art: SMF is compared against several established methods, which strengthens the paper's claims about the superiority of the proposed method.

**Weaknesses:**

- Lack of Theoretical Foundation: The paper could benefit from a deeper theoretical analysis of why SMF works better than existing methods. The underlying assumptions and mathematical properties of SMF need more exploration.
- Complexity and Scalability: The paper does not discuss the computational complexity of SMF or how it scales with larger datasets, which is crucial for practical applications.
- Limited Discussion on Hyperparameter Sensitivity: While the paper mentions hyperparameter tuning, there is limited discussion on how sensitive the model's performance is to these hyperparameters, which is important for reproducibility and practical use.
- Overfitting Concerns: The paper does not address potential overfitting issues, especially given the use of regularization terms in the loss function.
- Generalization to Other Domains: The paper primarily focuses on association data between two types of objects. It is unclear how well SMF generalizes to other types of data or more complex relationships.

**Questions:**

- How does SMF handle sparse data matrices, and what is its performance compared to other methods in such scenarios?
- Can the authors elaborate on any potential biases that might be introduced by the learned object similarities in SMF?
- What are the computational requirements for training SMF, and how does it compare to other methods in terms of training time and resource usage?
- How does SMF perform in dynamic environments where the association data changes over time, and is there any strategy to update the embeddings efficiently?
- Could the authors provide more insights into the choice of hyperparameters and their impact on the model's performance?

---

> ### Author Response · Authors · 2024-11-26
>
> We thank the thoughtful and constructive comments on our manuscript. Below, we carefully address your comments and show the revisions made to enhance the clarity and strength of the paper. We have also submitted a new version of the paper, where the changes are highlighted in red and AA stands for anonymous author.
> 1. **Lack of Theoretical Foundation.**
>
> In this revised version of the manuscript, we discuss SMF's theoretical foundations and how it works better than existing methods. We show this in the new section "Related works", and in "Self-Matrix Factorization" section.
>
> In "Related works", we wrote:
> > Manually curated similarities have proven useful for embedding learning, stemming from the fact that these similarities can themselves be used in recommender systems (Aggarwal et al., 2016). Sparse Linear Models (SLIM) (Ning & Karypis, 2011) are state-of-the-art recommender systems (Ferrari Dacrema et al., 2019) that rely on learning object similarities rather than embeddings. SLIM learns coefficients such that each object can be represented as a linear combination of other objects. This means that a new link between objects $i$ and $j$ is predicted only if objects similar to $i$ were originally linked with $j$. The coefficients used to reconstruct objects depend on the linear manifolds present in the data matrix $X$. In this way, new links are recommended to an object based on the links other objects belonging to the same linear manifold have. Although these similarities have demonstrated predictive power, they have not yet been used to inform embedding learning. In this work, we address this gap by proposing a framework that jointly learns object embeddings and object similarities, where the latter constrains the embedding space, resulting in richer representations.
>
> In "Self-Matrix Factorization", we modified the explanation of the model and Figure 1:
>
> >SMF learns two non-negative matrices $W \in \mathbb{R}^{n \times k}$ and $H \in \mathbb{R}^{k \times m}$, with $k<<(m \times n)$. Each matrix contains distinct low dimensional object embeddings, such that their product approximates the low-rank interaction data matrix $X \in \mathbb{R}^{n \times m}$:
> >$X \simeq WH.$
> >
> >While this model is not new, its novelty resides in the learning of the embeddings in $W$ to encode linear manifold information implicitly contained in the association data itself. Relying on the above mentioned assumption that objects lie on multiple linear low-dimensional manifolds embedded in high-dimensional space (Elhamifar & Vidal, 2013),, let us consider the situation depicted in Figure 1.a in which we have points in the 3-D space that are approximately localized onto 3 distinct linear manifolds. Rows of $X$ are represented as squares, triangles and circles, with triangles and squares lying on one-dimensional sub-space (red and brown lines) and circles lying on a two-dimensional sub-space (green plane). Let us focus on the three blue points of which $i$ and $p$ lie on the plane and $q$ on the red line. We assume that objects that belong to the same subspace, are more similar to each other than objects that reside in different subspaces. We would like these similarities to constrain the learning of the embeddings – that is, we would like the embedding for two objects that belong to the same subspace, to be more similar to each other than the embeddings of objects that reside in different subspaces. Thus, in the embedding space (2-dimensional, in Figure 1.b), object $i$ should be closer to object $p$ than to object $q$, mimicking their behavior in the high-dimensional space. Figure 1.b demonstrates the expected behavior of SMF-learned object embeddings. Points that belong to the same linear manifold in the high-dimensional space are projected into a lower-dimensional space, where they closely approximate one another.
> > By minimizing the loss function in equation 2, we approximate each interaction $X_{i,j}$ as $(W_{i,:}\cdot H_{:,j})$ (first term) as well as $\sum_s T_{i,s} (W_{i,:}\cdot W_{s,:}')X_{s,j}$ (second term). The first term enforces shared latent features between the rows and column objects, while the second term incorporates an explicit constraint for all the embeddings of the objects in the row of $X$. This second constraint is directly related to the similarity between object embeddings in $W$, so that the dot product between any pair $W_{i,:}$ and $W_{p,:}$ is informed by the linear manifolds in which objects $i$ and $p$ lies. Notably, SMF does not require prior knowledge of these manifolds; instead, it simultaneously learns the embeddings and the manifold structure, making it the first method to integrate these two processes.

---

> ### Author Response · Authors · 2024-11-26
>
> 2. **Complexity and Scalability**.
>
> Following the suggestion made by the reviewer, in the new version of the manuscript, we discussed the computational time complexity of SMF and how it scales to larger datasets. We added a paragraph in the section where we explain the model and created a new section in the Appendix.
> In "Self-Matrix Factorization", on page 5, we wrote:
> >From equations 5, 6, and 7, and assuming that  $k << n,m$, it follows that the time complexity of each iteration of SMF optimization algorithm is  $O(n^2 \cdot m)$. We discuss the complexity of SMF in the Appendix A.5, together with details about computational time and number of iterations.
> In Appendix, in section "A.5 Computational Time Complexity and Scalability ", we wrote:
> > We derived the computational time complexity of SMF optimization algorithm from equations 5,  6 and 7 that are used to check the convergence criterion and update $W$ and $H$, respectively. As each equation has a fixed number of matrix operations, it suffices to derive the asymptotic complexity of the most expensive one, which involves matrix multiplication. Specifically, the computation of the product between $T\circ(WW')$ and $X$ runs in time $O(n^2 \cdot m)$, where $n$ and $m$ are the dimensions of the data matrix $X \in \mathbb{R}^{n \times m}$. Other multiplications, such as $WH$, run in time $O(n\cdot k \cdot m)$, where $k$ is the dimension of the embedding space. Assuming that $k << n,m$, we have that the time complexity of each iteration of the SMF algorithm is $O(n^2 \cdot m)$.
> >
> > SMF iterations have the same time complexity as SEM's ($O(n^2 \cdot m)$), while NMF iterations are less expensive ($O(n \cdot k \cdot m)$). To illustrate the empirical running time of the three approaches,  we obtained the mean iteration time (in seconds)  in two datasets with different sizes (Movielens and ModCloth). This is shown in the first two columns of Table 8.  As SMF's multiplicative update rules involve more matrix multiplications than those of NMF and SEM, it is expected to have a higher execution time per iteration.
>
> Table 8. Mean iteration time and number of iterations
>
> |  Models | Movielens (s) |  ModCloth (s) | Movielens (it)  |ModCloth (it) |
> |--------------------------------|---------------------------------------|--------------------------------------|----------------------------------------|---------------------------------------|
> |  NMF                        | $0.01531$                             | $1.4287$                             | $1868.20$                              | $467$                                 |
> |  SEM                        | $0.4338$                              | $7.3948$                             | $244.20$                               | $292$                                 |
> |  SMF                        | $0.02737$                             | $7.4693$                             | $1427.73$                              | $1032$                                |
>
> > A fundamental variable for analyzing the overall computational time of the SMF algorithm is the total number of iterations until convergence. As it is not possible to derive it analytically, here we show only the empirical total number of iterations. The last two columns of Table 8 show the number of iterations that were necessary to achieve convergence for SMF, SMF, and SEM.
> >
> > The scalability of the SMF algorithm is inherently tied to that of NMF, as both share similar computational structures. Like NMF, SMF faces known limitations when applied to large-scale datasets, where the computational demands can hinder efficiency (Gan et al., 2021). This reflects a broader challenge in scaling matrix factorization techniques to accommodate increasingly larger datasets.
> >
> > All three methods require additional memory that exceeds the size of the input. Asymptotically, they have the same space complexity as the input $O(n \cdot m)$  when $n \leq m$. When $n >> m$, both SEM and SMF require space $O(n^2)$ to store the similarity matrix.

---

> ### Author Response · Authors · 2024-11-26
>
> 3. **Could the authors provide more insights into the choice of hyperparameters and their impact on the model's performance?**
>
> We ran new experiments to evaluate how the model performance is affected by different hyperparameters values. We gave insights about the hyperparameter's choice in the new subsection " SMF sensibility to hyperparameters", in section "4.1 Performance evaluation at predicting associations". We show the detailed results in Figure 4 in the Appendix.
> In subsection "SMF sensibility to hyperparameters" (page 7 and 8), we wrote:
>
> >**SMF sensibility to hyperparameters settings**. The SMF loss function proposed in Eq. 5 contains five hyperparameters. SMF demonstrates stable performance across a wide range of hyperparameter values, indicating that its practical application does not require extensive hyperparameter tuning. The parameter $\lambda_{se}$ controls the importance of the self-expressive term and we set it to 1 in all experiments in this paper. Figure 4 in the Appendix A.4 explores the effect other hyperparameters have on embedding learning by assessing the RMSE and AUPRC on the validation set using the MovieLens dataset. SMF is robust to the choice of the object embedding dimension $k$, achieving good performance even for low values of $k$. As it was also shown by other authors (Galeano et al., 2020), $\alpha$ value depends on the task. $\alpha$ should be set to a low value (closer to zero) when the objective is to accurately retrieve the numerical values of the associations, as in tasks focused on minimizing RMSE. Conversely, $\alpha$ should be set to a high value (closer to 1) when correctly identifying the associations themselves is more critical, as in tasks that optimize AUPRC. Additionally, this experiment shows that SMF is resilient to different values of the $\lambda_1$ and $\lambda_2$ regularization weights. Finally, performance remains consistent across the explored search space, with the only significant variations arising predictably from changes in $\alpha$.
>
> In Appendix A.4 Hyperparameter tunning, we added a Figure and the following paragraph:
>
> > Figure 4 shows the RMSE (orange) and AUPRC (blue) for different hyperparameter values. The red and grey lines divide the plot into different regions where $k$ (ranging from 4 to 16) and $\alpha$ (ranging from 0 to 1) are constant respectively. Note that within a region where $k$ is constant, there are five regions of constant $\alpha$. Within each of these regions, there are multiple values of the regularization weights in which $\lambda_2$ change after each consecutive point and $\lambda_1$ remains constant for 4 straight points. Both regularization weights range from zero to one and notably the lowest values of AUPRC correspond to regularization weights set to zero. Therefore, SMF generalizes better with regularized embeddings. Here, we illustrated SMF robustness across a wide range of hyperparameter values.
>
> 4. **Overfitting Concern**.
>
> To evaluate the impact of the regularization term for controlling overfitting, we showed how performance change with different values of  $\lambda_1$ and $\lambda_2$. This is discussed in the Appendix A.4 Hyperparameter tuning:
> >Both regularization weights range from zero to one and notably the lowest values of AUPRC correspond to regularization weights set to zero. Therefore, SMF generalizes better with regularized embeddings. Here, we illustrated SMF robustness across a wide range of hyperparameter values.
>
> 5. **Generalization to Other Domains**.
>
> We have added a paragraph to the Conclusion and Discussion section to clarify the types of data to which SMF can be applied:
> > Although SMF aims at learning from association data, it can be applied to any data represented as a nonnegative matrix. Furthermore, we expect it to be able to integrate extra information. For instance, to include additional measures of similarity between objects, one could add a term in the cost function that penalizes the difference between the learned similarity and the additional similarity measure.
>
> 6. **How does SMF handle sparse data matrices, and what is its performance compared to other methods in such scenarios?**.
>
> We would like to clarify that all matrices in our experiments are sparse. We show the density of the matrices in Table 1:
> | Datasets       | rows     | columns     | density      | NMF | HCCF | SMF |
> |----------------|----------|---------------|------------|--------|---------|-------|
> | Movielens    | $943$    |  $1682$      | $6.3\%$   | $10$ | $32$   | $10$ |
> |  Drug-SE        | $759$   | $994$          | $5\%$       | $10$ | $32$   | $10$ |
> |  ModCloth    | $5419$ |  $32089$    | $0.05\%$| $30$ | $32$   | $30$ |

---

> ### Author Response · Authors · 2024-11-26
>
> 7. **Can the authors elaborate on any potential biases that might be introduced by the learned object similarities in SMF?**.
>
> We added a paragraph on the potential biases in the Conclusion and Discussion section:
>
> > Like most machine learning models, SMF does not address biases in the training dataset. As a result, objects with more associations are likely to have higher prediction scores than those with fewer associations.
>
> 8. **What are the computational requirements for training SMF, and how does it compare to other methods in terms of training time and resource usage?**
>
> We created new sections in the Appendix to add this information.
> In Appendix A.2 Implementation Details, we describe specifications of the machine used to run our experiments:
>
> > Experiments were conducted on a machine equipped with two NVIDIA Quadro RTX 6000 GPUs (each with 24 GB of VRAM), an Intel Xeon Gold 6230 processor, and 192 GB of RAM. SMF, NMF, and SLIM were implemented in MATLAB R2023a, while HCCF was implemented in Python 3.6.12 using TensorFlow 1.14.0. CUDA version 11.8 was utilized to leverage GPU acceleration for HCCF training and evaluation.
>
> In addition, section "A.5 Computational Time Complexity and Scalability ", in Appendix, shows the empirical running time of SMF, empirical number of iterations, and time and space complexity.
>
> 9. **How does SMF perform in dynamic environments where the association data changes over time, and is there any strategy to update the embeddings efficiently?**
>
> We answered this question in the new version of Conclusion and Discussion:
>
> > Another limitation of SMF is its inefficiency in handling dynamic datasets. When association data changes, the model must be retrained to ensure that the embeddings reflect the updated state of the objects.

---

### Meta-Review · Area_Chair_ocox · 2024-12-20

**Metareview:**

The paper received three negative ratings, with all reviewers inclined to reject it. It introduces SMF but lacks a solid theoretical foundation, offering insufficient explanation for why SMF outperforms existing methods. The assumptions and mathematical properties of SMF need further analysis, and there is no discussion of its computational complexity, scalability, or sensitivity to hyperparameters and overfitting. The generalization of SMF to other domains is also unexplored.  The paper lacks a related works section, making it unclear how SMF compares to other matrix factorization methods, especially those with similar self-expressive constraints. Figures are generic and do not effectively highlight SMF's unique aspects, while the small datasets used limit the results' generalizability. Although SMF shows slight improvements over NMF, the results are not convincing due to limited experiment design and outdated methods. The claim about deriving object similarities directly from the data matrix is unclear, and the experimental section is overly simplistic. Despite the authors' efforts to address these issues, the reviewers maintain their negative ratings. Therefore, the Area Chair recommends rejection.

**Additional Comments On Reviewer Discussion:**

Despite the authors' efforts to address these issues, the reviewers maintain their negative ratings.

---

### Decision · Program_Chairs · 2025-01-22

Reject